# JUDO: A Juxtaposed Domain-Oriented Multimodal Reasoner for Industrial Anomaly QA

**Hyunju Kang**[1,*]   **Woohyun Lee**[1,*]   **Jaewon Kim**[2]   **Hogun Park**[1,†]
[1]Sungkyunkwan University   [2]Seoul National University
{neutor, woodavid31}@skku.edu
{neoblue95}@snu.ac.kr, {hogunpark}@skku.edu

## Abstract

Industrial anomaly detection has been significantly advanced by Large Multimodal Models (LMMs), enabling diverse human instructions beyond detection, particularly through visually grounded reasoning for better image understanding. However, LMMs lack domain-specific knowledge, which limits their ability to generate accurate responses in complex industrial scenarios. In this work, we present JUDO, Juxtaposed Domain-Oriented Multimodal Reasoner, a framework that efficiently incorporates domain knowledge and context in visual and textual reasoning. Through visual reasoning, our model segments the defect region by juxtaposing query images with normal images as visual domain context, enabling a fine-grained visual comparative inspection. Furthermore, we inject domain knowledge through supervised fine-tuning (SFT) to enhance context understanding and subsequently guide domain reasoning through reinforcement learning (GRPO) with tailored rewards, opting for a domain-oriented reasoning process. Experimental results demonstrate that JUDO achieves superior performance on the MMAD benchmark, surpassing models such as Qwen2.5-VL-7B and GPT-4o. These results highlight the importance of enhancing domain knowledge and context for effective reasoning in anomaly understanding.

## 1 Introduction

Visual anomaly detection (Jiang et al., 2025; Xu et al., 2025), which aims to identify anomaly patterns from images in industrial environments, has evolved significantly with the emergence of Large Multimodal Models (LMMs) (Hurst et al., 2024; Bai et al., 2025; Wang et al., 2025). Although most anomaly patterns are typically domain-specific, current LMMs have demonstrated reasonable generalization capabilities in complex industrial environments, as shown by the MMAD benchmark (Jiang et al., 2025). Building upon this foundation, recent models such as AnomalyR1 (Chao et al., 2025) and OmniAD (Zhao et al., 2025) have employed Group Relative Policy Optimization (GRPO) (Shao et al., 2024) to further advance multimodal anomaly detection. Specifically, AnomalyR1 is the first to introduce GRPO training into anomaly detection tasks, while OmniAD advances this approach by integrating anomaly segmentation to enable visual grounded reasoning.

Despite these advances, current GRPO-based models (Chao et al., 2025; Zhao et al., 2025) primarily optimize instruction-response matching, overlooking the incorporation of domain knowledge essential for generalizable defect reasoning. Such domain knowledge typically encompasses characteristics of normal and defect samples expressed in textual form, including their definitions, causes, and consequences, which serve as essential priors for reliable defect analysis. In addition, normal images serve as visual context, providing references for distinguishing normality from anomaly. However, such knowledge and visual context are often highly specialized and rarely encountered during LMM pretraining, making reasoning based solely on their inherent knowledge insufficient. Recent studies (Jiang et al., 2025) have demonstrated that providing external knowledge or normal samples (Zhao et al., 2026) as context in prompts during inference can alleviate this limitation. However, providing knowledge and context only at inference time remains limited when LMMs lack

---

[*]Equal contribution.
[†]Corresponding author.

sufficient internal knowledge, as they may become overly dependent on external context, leading to misaligned responses that prioritize contextual plausibility over accuracy. This limitation ultimately stems from insufficient internalized domain knowledge and contextual understanding, hindering reliable and accurate domain-oriented reasoning.

To address this challenge, we propose **JUDO**, **Ju**xtaposed **D**omain-**O**riented Multimodal Reasoner, the first approach that systematically internalizes domain knowledge for industrial anomaly detection through learning. Unlike prior GRPO-based models that employ post-training without domain alignment, JUDO unifies domain understanding across visual grounding and textual reasoning. Our framework begins in Stage 1 by establishing visual reasoning through juxtaposed segmentation learning by comparing normal and defect images. This approach leverages the underexplored potential of normal samples, which typically serve only as optional context in inference, while JUDO incorporates it in a core reasoning context during training. In Stage 2, we enhance domain-oriented textual reasoning by injecting domain knowledge into model parameters, in contrast to prior work (Jiang et al., 2025) that introduces textual domain knowledge externally via prompts. This stage builds foundational domain knowledge for industrial anomaly reasoning, yielding more reliable domain-aligned reasoning. Finally, Stage 3 unifies visual grounding and domain semantics through reinforcement learning (GRPO) (Shao et al., 2024), using rewards for domain reasoning, segmentation, choice accuracy, and structural alignment rewards to ensure that the model produces accurate domain-aware anomaly understanding.

Extensive experiments on the industrial anomaly detection benchmark MMAD (Jiang et al., 2025) demonstrate the effectiveness of our approach. JUDO achieves superior performance, showing that jointly learning domain knowledge and visually grounded reasoning substantially improves industrial anomaly understanding. Beyond accuracy gains, JUDO enhances reliability and explainability by grounding predictions in localized defect regions through anomaly segmentation and aligning reasoning with domain-specific knowledge. This unified training enables interpretable and visually verifiable decision-making while maintaining strong performance across diverse defect-related tasks. The implementation of JUDO is publicly available.[1]

We summarize our contributions as follows:

- We propose JUDO, the first approach to systematically internalize domain knowledge and context into both visual and textual reasoning for industrial anomaly understanding.

- We present a unified framework that builds domain understanding through visual segmentation and textual knowledge internalization. We integrate these capabilities via reinforcement learning with our domain-aligned reward designs.

- Through comprehensive experiments, we demonstrate that JUDO achieves superior performance over prior GRPO-based models, while improving explainability through anomaly-grounded segmentation and enhancing reliability via domain-aligned reasoning for real-world industrial scenarios.

## 2 RELATED WORK

### 2.1 LARGE MULTIMODAL MODELS (LMMs)

Recent Large Multimodal Models (LMMs) have advanced visual understanding through high-quality instruction tuning (Hurst et al., 2024; Bai et al., 2025; Chen et al., 2024; Kimi Team et al., 2025; Comanici et al., 2025), stronger cross-modal architectures (Wang et al., 2025; Agrawal et al., 2024b), and more sophisticated training pipelines (Xiaomi et al., 2025; Deitke et al., 2025). Most multimodal models now support multi-image inputs (Xiaomi et al., 2025), with GPT-4o (Hurst et al., 2024), GPT-5 (Singh et al., 2025), the Gemini-2.5 model (Comanici et al., 2025), and InternVL3.5 (Wang et al., 2025) demonstrating improved reasoning performance through cross-image comparison. However, the majority of these models (Hurst et al., 2024; Comanici et al., 2025; Bai et al., 2025; Wang et al., 2025) remain optimized for common question–answering tasks grounded in general knowledge. Consequently, their reasoning performance drops noticeably in specialized application domains, where domain knowledge is critical.

---

[1]Code available at https://github.com/woodavid31/JUDO

## 2.2 INCORPORATING DOMAIN KNOWLEDGE INTO LMMS

To address the demand for specialized knowledge for LMMs (Song et al., 2025), specifically in fields such as finance (Qian et al., 2025), biomedicine (Liu et al., 2025b), education (Agrawal et al., 2024a), and materials (Prabhakar et al., 2025), researchers have actively studied how to integrate domain-specific expertise into LMMs. Most approaches fall into two categories: dynamic injection and learning-based integration through pretraining or fine-tuning. Dynamic injection-based methods provide the external knowledge in a similar way to RAG (Zhao et al., 2026) at inference time without additional training; their effectiveness is highly dependent on retrieval quality. In contrast, learning-based methods encode the domain knowledge into the model's parameters. However, this line of work remains underexplored for industrial anomaly understanding.

## 2.3 INDUSTRIAL ANOMALY DETECTION AND LMMS

Visual anomaly detection in industrial settings has evolved significantly with deep learning. Early and still widely used methods are often unsupervised, focusing on learning a model of normal data and identifying deviations (Li et al., 2025; Bergmann et al., 2021). These include reconstruction-based methods using autoencoders or GANs, where high reconstruction error signals an anomaly (An & Cho, 2015; Schlegl et al., 2017), and embedding-based methods, which map normal samples to a tight cluster in a feature space (Roth et al., 2022; Defard et al., 2021). While effective for detection and localization on benchmarks like MVTec AD (Bergmann et al., 2019), these approaches typically do not provide defect analysis for their predictions.

The advent of LMMs has introduced a new paradigm that enables comprehensive defect analysis. Rather than simply detecting anomalies, LMMs can now respond to diverse analytical queries about defect characteristics—such as identifying defect types, describing their visual appearance, and analyzing their potential consequences. To facilitate research in this area, a MMAD benchmark (Jiang et al., 2025) has been presented, providing a suite for evaluating an LMM's reasoning abilities on industrial anomaly problems. Meanwhile, several works have recently applied LMMs to this challenge. AnomalyGPT (Gu et al., 2024) was an early method using LMMs for zero-shot anomaly detection and generating descriptive reports. More recently, a few methods employ reinforcement learning to improve reasoning quality. AnomalyR1 (Chao et al., 2025) incorporates Group Relative Policy Optimization (GRPO) (Shao et al., 2024) to refine its reasoning. Furthermore, OmniAD (Zhao et al., 2025) unifies the anomaly segmentation and anomaly reasoning problem for fine-grained defect understanding.

## 3 METHOD

Our proposed model, **JUDO**, is a domain-oriented multimodal reasoner for industrial anomaly understanding. The main claim of JUDO is that domain knowledge and contextual information fundamentally matter in industrial anomaly understanding, yet this approach remains underexplored and non-trivial. In this direction, our core contribution is a unified learning objective in which comparative visual reasoning and domain semantics fuse together into the domain-aligned reasoning process, described in Figure 1. The first stage introduces a comparative reasoning ability between query and normal images for patch-level segmentation, then the second stage internalizes textual domain knowledge into model parameters, and the final stage unifies the visual grounding and domain semantics using GRPO-based optimization with tailored reward designs. The following subsections detail each training stage, with the corresponding dataset construction process described in Appendix A.

## 3.1 STAGE 1: LEARNING ANOMALY SEGMENTATION-BASED JUXTAPOSED REASONING

For reliable and fine-grained visual reasoning, we employ anomaly segmentation to pinpoint defect regions and integrate this capability into the reasoning process. To encompass proper domain visual context, we propose the juxtaposed reasoning paradigm that explicitly compares defective images against their normal samples during training. While most existing segmentation methods achieve reasonable performance by directly predicting the target regions, the lack of clear reasoning criteria for anomaly judgment limits segmentation performance in defect detection. Additionally, although

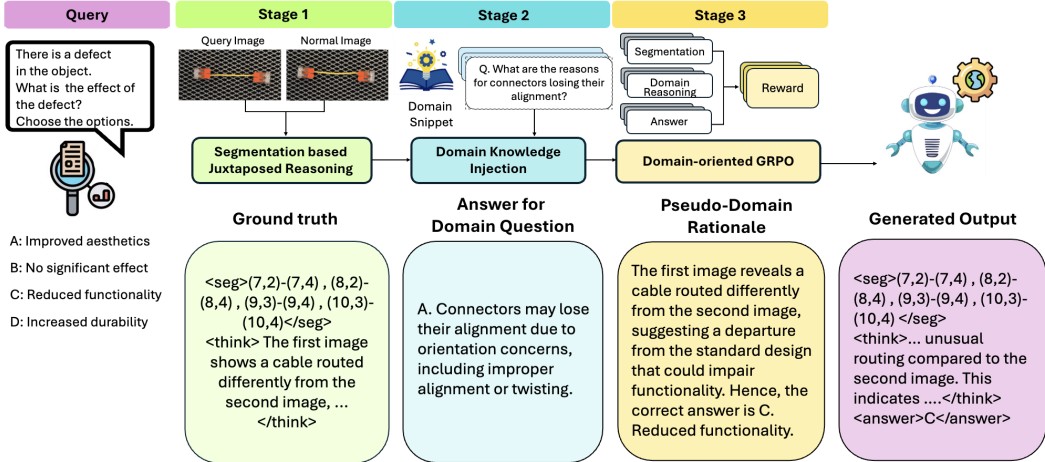

Figure 1: Overview of JUDO (Juxtaposed Domain-Oriented Multimodal Reasoner). The framework progresses in three stages: (1) Stage 1: Learning anomaly segmentation-based juxtaposed reasoning, (2) Stage 2: Domain-knowledge injection, and (3) Stage 3: Domain-oriented group relative policy optimization. Through the progressive stages, we incorporate the domain knowledge and context into LMMs for reliable and robust anomaly analysis.

using normal images as contextual reference has become common practice during inference (Jiang et al., 2025), how to efficiently harness this comparative context to enable explicit reasoning of defects during training has yet to be addressed. Our comparative training paradigm shifts the objective from a simple pattern memorization to a fine-grained juxtaposed reasoning, enabling the model to internalize a more generalized and fundamental notion of *normality* as a baseline for anomaly understanding.

Inspired by text-based segmentation approaches, such as Text4Seg (Lan et al., 2025; Zhao et al., 2025), the model is trained to output the coordinates of anomalous patches within a $16 \times 16$ grid through SFT (Ouyang et al., 2022). The instruction is given to describe the anomaly region and explain the visual evidence by comparing query images with normal samples as juxtaposed reasoning. For instance, a defect region is represented as a textual sequence such as (11,12)-(11,14), (12,11), pinpointing specific patches that deviate from the norm, including juxtaposed explanations. This format compels the model to perform a direct, patch-level juxtaposition, moving beyond a general comparison to a fine-grained recognition of differences against the normal template. This spatial grounding is crucial, as it forces the reasoning process to be tied to specific visual evidence, making the subsequent textual explanation more accurate and reliable.

**Dataset Construction.** We construct a training dataset for juxtaposed fine-grained anomaly segmentation. We utilize only one image from each defect category in the MMAD dataset, and additionally, incorporate the REAL-IAD (Wang et al., 2024) dataset for more generalized segmentation capability. As illustrated in Figure 6 of the Appendix, each anomalous query image is paired with a randomly sampled normal template from the same object category. The model is then trained to generate a dual-part response from this pairing: first, a textual sequence of patch coordinates between `<seg></seg>` tags that serves to ground the explanation by identifying the anomalous region, and second, a corresponding comparative explanation within `<think></think>` tags, which is synthetically generated to describe the defect by contrasting the two images, given the anomalous region bounded by a red line and the defect type.

### 3.2 STAGE 2: DOMAIN-KNOWLEDGE INJECTION

While Stage 1 equips the model with comparative reasoning ability using normal images as the domain context, it primarily focuses on visual reasoning processes. However, the model still lacks sufficient domain knowledge for industrial anomaly problems. As a result, the model struggles to deliver accurate textual reasoning and often fails to derive correct solutions without proper foun-

dational knowledge. To address this limitation, we inject the domain knowledge (Mecklenburg et al., 2024) by constructing domain QA datasets and employing supervised fine-tuning (SFT). This approach differs from conventional supervised learning that directly trains on correct answers to queries in the MMAD dataset, as it focuses on supervised learning of domain knowledge that may not directly correspond to specific instructions but provides foundational knowledge essential for solving related problems. Through supervised fine-tuning on the domain QA dataset, the model acquires a more generalizable understanding of how domain knowledge applies across different object categories and defect types. This enhanced understanding forms the basis for Stage 3, where reinforcement alignment further refines the precision and reliability of domain reasoning.

**Dataset Construction.** We generate question–answer pairs using the raw domain snippets provided in MMAD (Jiang et al., 2025). Each snippet contains an unstructured textual description of the characteristics of an object category and its associated defect types, and serves as a textual knowledge source for constructing the Stage 2 dataset (see Table 4 in the Appendix). Specifically, we prompt GPT-4o to reflect inspection knowledge from the unstructured domain snippet into structured QA pairs —for example, questions related to defect criteria, functional implications (e.g., "What criteria indicate that the fabric border is defective?" or "Why is detecting loose threads important for product reliability?"). These questions are derived purely from the textual information in the snippet and are not tied to any specific anomalous image sample. To improve robustness, each QA is further paraphrased into multiple semantically consistent but lexically diverse variants. We show a more detailed data construction process in Section A. Finally, every QA instance is paired with a normal image from the corresponding object category, grounding the textual domain knowledge in its object-level visual context and naturally prompting the model to recall relevant domain knowledge during reasoning. We provide examples of the generated QA pairs in Section B.2.

## 3.3 STAGE 3: DOMAIN-ORIENTED GROUP RELATIVE POLICY OPTIMIZATION

The final stage of our framework, Stage 3, is designed to elevate the model's capabilities in the visual and textual domain-oriented reasoning, mainly obtained from prior stages. While Stage 1 builds a foundation in visual juxtaposed reasoning and Stage 2 internalizes textual domain knowledge, these skills are not yet integrated. Thus, we employ Group Relative Policy Optimization (GRPO) (Shao et al., 2024), which is a reinforcement learning method that optimizes policies by comparing relative performance within groups of trajectories for better generalization. We design GRPO policies to align the model's behavior, ensuring that the final output is a seamless integration of accurate visual grounding and deep domain understanding. This alignment is guided by multi-faceted reward functions that provide a comprehensive feedback signal, composed of three primary components. We explain each policy with respect to reward functions as follows.

**Domain Reasoning Reward.** The primary objective of the Domain Reasoning Reward is to guide the reasoning process presented within the `<think></think>` tags toward aligning with the domain-oriented reasoning process, specifically queries targeting the MMAD benchmark. Since there is no ground-truth for each instruction encompassing domain knowledge and context for MMAD datasets, we generate the *pseudo-domain rationale* for the 1-shot train split using GPT-4o, providing full contexts such as the query, correct answer, images including normal reference, juxtaposed reasoning from Stage 1 pipeline, and relevant domain knowledge. We treat this generated pseudo-domain rationale as the semantic target and design the policy to guide domain-oriented direction. Specifically, we define the reward, $R_{domain}$, using the cosine similarity between the embedding vectors of the model's generated reasoning as $E_{gen}$ and a pseudo-domain rationale denoted as $E_{pdomain}$ using the representation space of **all-MiniLM-L6-v2** SentenceTransformer (Reimers & Gurevych, 2019) model, which effectively encodes the semantic meaning of one or multiple sentences, denoted as $\phi(\cdot)$. This reward mechanism is formalized as:

$$R_{\text{domain}} = \lambda \cdot \frac{\phi(E_{\text{gen}}) \cdot \phi(E_{\text{pdomain}})}{\|\phi(E_{\text{gen}})\| \, \|\phi(E_{\text{pdomain}})\|}. \tag{1}$$

The domain reasoning reward leverages semantic similarity to align the model's reasoning with a reference rationale that is grounded strictly in the provided evidence. The pseudo-domain rationale reorganizes the inputs without introducing new knowledge, ensuring that GPT-4o serves only as an evidence structurer. This reward differs from typical GRPO settings that emphasize answer correct-

ness or output format, since it directly encourages consistent domain-oriented reasoning patterns while remaining flexible to the phrasing of the generated explanation.

**Segmentation Reward.** This reward evaluates the spatial precision of the anomaly patch coordinates generated within the `<seg></seg>` tags through F1 score calculation (Zhao et al., 2025), reinforcing the skill developed in Stage 1. A reward of 0 is immediately assigned for any improperly formatted coordinate strings. For validly formatted outputs, the reward function compares the set of predicted grid cells (P) against the ground-truth set ($P_G$) using the following piecewise function:

$$R_{seg} = \begin{cases} 1.0 & \text{if } P = \emptyset \text{ and } P_G = \emptyset, \\ 0.2 + 0.8 \cdot \text{F1}(P, P_G) & \text{if } P \neq \emptyset \text{ and } P_G \neq \emptyset, \\ 0.0 & \text{otherwise.} \end{cases} \quad (2)$$

This formulation incentivizes not only high-overlap localization but also correct format adherence and accurate identification of anomaly-free instances, penalizing cases where only one set is empty.

**Choice and Structural Alignment Reward.** This composite reward ensures correct, well-structured, and logically sound output through three components (Shao et al., 2024): (1) **Choice Reward** rewards correct multiple-choice selection within `<answer></answer>` tags to promote accurate decision-making; (2) **Format Reward** ensures adherence to the required format as `<seg>...<think>...<answer>` structure, ensuring the structure of responses is consistent and parsable; and (3) **Reasoning Structure Reward** encourages both the conclusive answer within the reasoning and final answer to be correct while penalizing answer choices mentioned in the first half of reasoning text, preventing premature commitment.

## 4 EXPERIMENTS

### 4.1 EXPERIMENTAL SETUP

**Datasets and Benchmarks.** We evaluate our approach on the MMAD benchmarks (Jiang et al., 2025), which integrate four datasets: MVTec AD (Bergmann et al., 2019), MVTec LOCO (Bergmann et al., 2022), VisA (Zou et al., 2022), and GoodsAD (Zhang et al., 2024) datasets, with multiple-choice QA covering seven key subtasks. We report averaged accuracy as the evaluation metric, computed as the ratio of correct predictions to the total number of predictions. For Stage 1, we sample 10 instances from each category in the Real-IAD dataset (Wang et al., 2024) together with one instance per category from MMAD, yielding 1.4k and 293 images, respectively. In Stage 2, we construct a domain-specific QA corpus by leveraging the MMAD domain knowledge JSON files. For each category, we generate 30 unique questions and augment them with two paraphrased variants, resulting in approximately 13k QA pairs. Finally, Stage 3 applies GRPO using a sparse sampling strategy, where only one training instance per category from MMAD—identical to the samples used in Stage 1—is utilized, yielding 1.4k QA pairs for reinforcement alignment.

**Implementation Details.** Our framework is built on PyTorch 2.5.1 with Hugging Face Transformers and the TRL GRPO trainer. We use Qwen2.5-VL-7B as the base model, initialized from a vision–language pre-trained checkpoint. Training runs on a single node with 4×NVIDIA H200 GPUs using `torchrun` and DeepSpeed ZeRO-3. For Stage 1 and Stage 2, we train for 8 and 2 epochs, respectively, using learning rates of $1 \times 10^{-6}$ and $5 \times 10^{-7}$ in `bf16` precision. For GRPO, we use 16 generations per prompt, a batch size of 8, and train for 14 epochs in `bf16` precision. Additionally, we set the domain reasoning reward coefficient $\lambda = 0.1$ to avoid over-emphasizing semantic similarity to the pseudo-domain rationale, ensuring that it acts as a soft alignment signal rather than dominating the multi-reward optimization.

**Baselines and Inference Settings.** We benchmark JUDO against leading general-purpose models such as Qwen2.5-VL, InternVL3.5, and various commercial models. Our primary industrial specialized baseline is AnomalyR1 (Chao et al., 2025); another recent method, OmniAD (Zhao et al., 2025), was not included because its public codebase was unavailable as of February 2026. For consistency with JUDO's architecture, we re-implemented the AnomalyR1 approach on a Qwen2.5-VL 7B model using the author's provided code and our sampled dataset. The result is referred to as the AnomalyR1 in Table 1 and the Base GRPO model in our ablation study in Section 4.3. All models are evaluated under a strict 1-shot inference protocol in the same way as MMAD, using both a query image and a normal template as input to ensure a fair comparison.

Table 1: Performance comparison of both commercial and open-source LMMs in MMAD with the standard 1-shot setting. Anomaly Discrimination uses the average accuracy (%) across the normal and abnormal categories. The best scores are highlighted in bold, while the second best are underlined. * indicates the best performance among the open-source models.

| Model | Scale | Anomaly Discrimination | Defect | | | | Object | | Average |
|---|---|---|---|---|---|---|---|---|---|
| | | | Classification | Localization | Description | Analysis | Classification | Analysis | |
| Random Chance | - | 50.00 | 25.00 | 25.00 | 25.00 | 25.00 | 25.00 | 25.00 | 28.57 |
| Human (expert) | - | 95.24 | 75.00 | 92.31 | 83.33 | 94.20 | 86.11 | 80.37 | 86.65 |
| Human (ordinary) | - | 86.90 | 66.25 | 85.58 | 71.25 | 81.52 | 89.58 | 69.72 | 78.69 |
| Claude-3.5-sonnet | - | 60.14 | 60.14 | 48.81 | 67.13 | 79.11 | 85.19 | 79.83 | 68.36 |
| Gemini-1.5-pro | - | 68.63 | 60.12 | 58.56 | 70.38 | 82.46 | 89.20 | 82.25 | 73.09 |
| Gemini-2.5-pro | - | 83.07 | 73.86 | 67.20 | 79.97 | 86.27 | 94.88 | 83.08 | 81.19 |
| Gemini-2.5-flash | - | **93.38** | 69.88 | 63.30 | 76.41 | 81.57 | 94.04 | 82.00 | 80.08 |
| GPT-4o | - | 68.63 | 65.80 | 55.62 | 73.21 | 83.41 | **94.98** | 82.80 | 74.92 |
| GPT-5-mini | - | 64.10 | 67.35 | 69.07 | 79.02 | 86.72 | 93.96 | 83.37 | 77.65 |
| Qwen2.5-VL | 7B | 71.39 | 54.35 | 61.17 | 65.81 | 79.32 | 91.44 | 84.43 | 72.56 |
| LLaVA-OneVision | 7B | 51.77 | 46.13 | 41.85 | 62.19 | 69.73 | 90.31 | 80.93 | 63.27 |
| InternVL3.5 | 8B | 67.50 | 49.37 | 57.90 | 58.07 | 77.66 | 72.01 | 81.11 | 66.30 |
| Kimi-VL-A3B | 16B | 72.93* | 53.49 | 59.66 | 72.39 | 81.74 | 91.91 | 85.89 | 74.00 |
| MiMo-VL | 8B | 54.45 | 59.56 | 60.77 | 71.50 | 78.48 | 90.56 | 81.60 | 70.99 |
| AnomalyR1 | 7B | 60.93 | 64.81 | 70.72 | 79.06 | 85.52 | 93.12 | 86.91 | 77.29 |
| JUDO | 7B | 65.04 | **74.74*** | **73.01*** | **84.56*** | **89.41*** | 94.04* | **87.58*** | **81.20*** |

## 4.2 EXPERIMENTAL RESULTS

Table 1 presents a comparison of our model with both general-purpose LMMs and the anomaly-focused baseline, such as AnomalyR1. JUDO achieves strong performance across the MMAD seven subtasks, reaching the highest overall averaged accuracy of 81.20%. Advantages of JUDO are particularly evident across the four defect-related subtasks: classification, localization, description, and analysis. Since these tasks critically depend on domain-specific knowledge, the performance gains highlight the effectiveness of incorporating domain knowledge in reasoning through our progressive stages. Specifically, Stage 1 establishes a strong foundation by using juxtaposed segmentation training to foster fine-grained comparative analysis, which directly boosts defect localization to 73.01% (from 61.17% for the base Qwen). This visual grounding is then enriched in Stage 2, where the integration of domain-specific knowledge provides the conceptual basis needed for accurate explanations, as seen in the 84.56% and 89.41% accuracy on the defect description and analysis, respectively. Finally, Stage 3 is critical for merging these capabilities. The multi-reward GRPO training ensures the model's final output is a coherent synthesis of juxtaposed visual evidence and textual expertise. This integrated process allows JUDO to not only locate defects precisely but also to classify, describe, and analyze them with a domain-aligned thinking process, leading to more robust and reliable reasoning.

However, our JUDO does not achieve superior performance in anomaly discrimination tasks, even with rich domain-specific foundational knowledge. Similarly, other models trained with a GRPO framework, such as AnomalyR1, exhibit a trade-off in raw binary detection accuracy. This is evident in the results: AnomalyR1 achieves 60.93% accuracy, while JUDO shows only a modest improvement at 65.04%. Both remain below the 72.93% and 71.39% reached by recent LLMs such as Kimi-VL, as well as the base model Qwen2.5-VL, respectively. We attribute Kimi-VL's higher performance to its more advanced visual encoder, although it still lacks domain-specialized knowledge, as reflected in its relatively low performance on defect-related tasks. Commercial multimodal models such as Gemini-2.5-Pro and Gemini-2.5-Flash also show strong performance on anomaly discrimination, likely due to their highly capable vision encoders.

Even in the base model Qwen2.5-VL, which is not equipped with a stronger vision encoder, we observe a similar performance drop in the base model Qwen2.5-VL when switching from direct answering to reasoning mode. This observation suggests that the degradation is not solely attributable to encoder capacity but is instead attributable to the introduction of reasoning itself. Notably, the drop is more pronounced in relatively simple anomaly discrimination settings, a phenomenon that has also been observed in recent studies on related tasks (Liu et al., 2025a; Yang et al., 2025). We provide a more detailed analysis of this phenomenon in Section 4.4, where we examine the impact of our multi-staged learning framework on the anomaly discrimination task.

Table 2: Ablation study of different methods. We denote Stage 1 as SegJux, Stage 2 as DomInj, and the full Stage 3 as GRPO$^{dom}$, where the domain reasoning reward is used. The segmentation reward is only used in methods that go through Stage 1 training. Average accuracy (%) is reported.

| Method | Average |
|---|---|
| Qwen2.5-VL-7B | 72.56 |
| + GRPO | 77.29 |
| + GRPO + RAG | 76.29 |
| + GRPO + DomInj | 79.82 |
| + GRPO + SegJux + DomInj | 80.35 |
| + GRPO$^{dom}$ + SegJux + DomInj (JUDO) | **81.20** |

Except for the relatively simple anomaly discrimination task, the performance of both recent open- and closed-source LMMs on domain-level defect reasoning remains overall lower than JUDO's. This contrast highlights a key distinction: while large commercial LMMs excel at broad visual pattern recognition, they do not have sufficient defect semantics or comparative industrial cues in a way that generalizes across classification, localization, description, and analysis. In contrast, despite being built on a smaller open-source backbone, JUDO consistently achieves higher defect-reasoning accuracy because its training explicitly incorporates domain knowledge and visual context. These results demonstrate that domain-aligned training can surpass much larger commercial systems when the task requires specialized industrial reasoning rather than generic visual understanding.

## 4.3 ABLATION STUDY

We conduct an ablation study to systematically evaluate the contribution of each component in the JUDO framework, with results summarized in Table 2. The study begins with the baseline Qwen2.5-VL-7B model, which achieves an average accuracy of 72.56%. We note that this result is obtained without using the Chain-of-Thought process, not requiring the reasoning process generally in <think> tags, but outputting the answer directly, since it is not trained to learn the proper reasoning, yet. Applying a standard GRPO training stage improves general instruction-following and raises the accuracy to 77.29% in the equivalent of AnomalyR1. To evaluate the effectiveness of domain knowledge injection, we employ RAG, which dynamically provides external information upon the model trained on vanilla GRPO, referring to the model as (+ GRPO + RAG). The way of employing RAG is followed in the MMAD (Jiang et al., 2025). While RAG-based approaches have been claimed to often improve the performance, the external context attached to our trained reasoning model acted in the negative direction rather than bringing in positive gain. In contrast, the impact of domain injection was remarkably effective, resulting in an improvement of almost 5% gain.

The effectiveness of JUDO's learning-based approach becomes evident in the subsequent steps, as shown in the result. The most significant performance leap comes from Stage 2's Domain Injection (+ GRPO + DomInj), which internalizes domain knowledge through supervised fine-tuning and increases accuracy to 79.82%. This result is substantially higher than the RAG-based method, demonstrating the superiority of integrating textual knowledge directly into the model's parameters. Building on this, the addition of Stage 1's juxtaposed segmentation training (+ GRPO + SegJux + DomInj) brings the accuracy to 80.35%, highlighting the value of grounding textual reasoning in fine-grained juxtaposed visual analysis. The final refinement, which uses the complete Stage 3

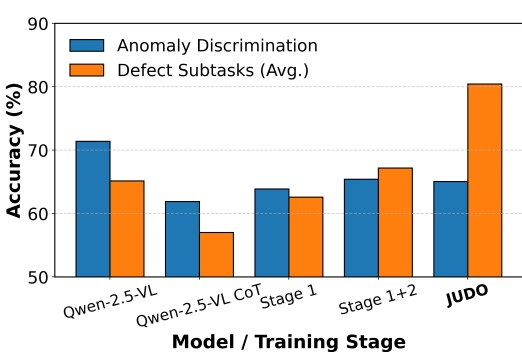

Figure 2: Performance across different stages.

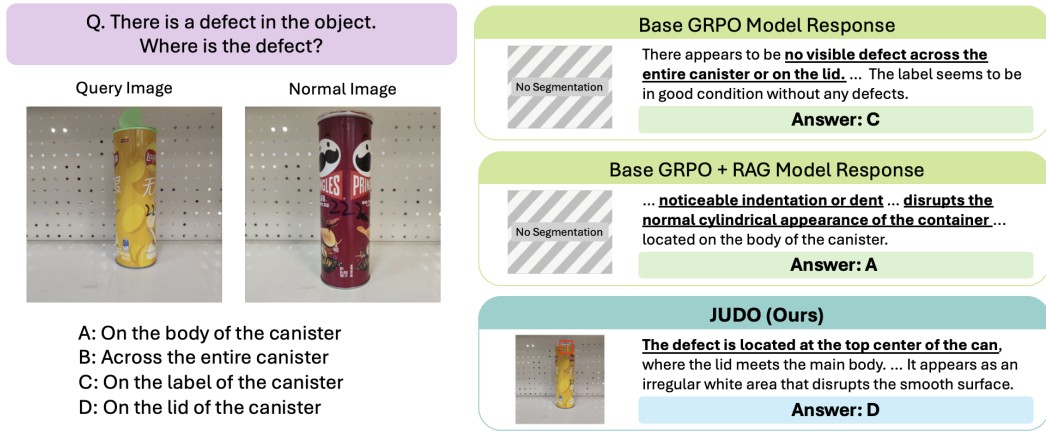

Figure 3: Response comparison between Base GRPO, Base GRPO + RAG and JUDO. The anomalous region in the query image is highlighted in green, while the segmentation output from JUDO is represented as red patches in the image next to JUDO's response.

GRPO with a specialized domain reasoning reward (+ GRPO$^{dom}$), achieves the peak performance of 81.20%. This confirms that each stage of the JUDO framework provides a meaningful and incremental benefit, leading to its final state-of-the-art performance.

## 4.4 DISCUSSION OF MULTI-STAGED LEARNING FRAMEWORK

Figure 2 presents the progression of anomaly discrimination accuracy and averaged accuracy of defect subtasks (classification, localization, description, and analysis) across the key checkpoints of our pipeline: the Qwen2.5-VL baseline, its Chain-of-Thought (CoT) variant, and each JUDO stage. The figure shows that the most significant performance shift occurs before any of JUDO's optimization takes place. When the model switches from direct answering to CoT reasoning, anomaly discrimination accuracy drops sharply (71.39% → 61.90%), and the averaged accuracy over the four defect subtasks decreases from 65.15% to 57.00%. This confirms that the initial degradation stems from a reasoning-mode behavioral change rather than from the design of our multi-stage training. The drop in anomaly discrimination in particular is consistent with a recent finding (Liu et al., 2025a) that explicit verbal reasoning can impair performance on perception-heavy tasks by interfering with rapid, pattern-based visual judgments, a phenomenon analogous to verbal overshadowing. After entering the pipeline, anomaly-detection performance remains largely stable, with only a small fluctuation from 65.42% (Stage 1+2) to 65.04% in the final model, indicating that catastrophic forgetting is present but limited.

In contrast, the progressive optimization produces substantial and consistent gains across the four defect subtasks. As shown in Figure 2, accuracy increases from 62.58% after Stage 1 to 67.18% after Stage 1+2, and ultimately reaches 80.43% in the full model. This pattern demonstrates that the GRPO-based domain alignment strengthens domain-aware classification, localization, description, and analysis, even as low-level anomaly discrimination remains stable. These results highlight that JUDO's multi-stage design primarily enhances domain-oriented reasoning capabilities while preserving binary anomaly detection performance. Overall, despite a minor drop in vision-oriented anomaly discrimination, JUDO significantly improves performance on reasoning-intensive defect subtasks, while additionally improving reliability and explainability through localized defect grounding and domain-aligned reasoning.

## 4.5 QUALITATIVE ANALYSIS

Figure 3 provides a qualitative comparison that highlights the limitations of baseline models and the effectiveness of JUDO's integrated reasoning. The Base GRPO model exhibits a critical failure, as it is unable to detect the defect and states there is "no visible defect," while paradoxically offering

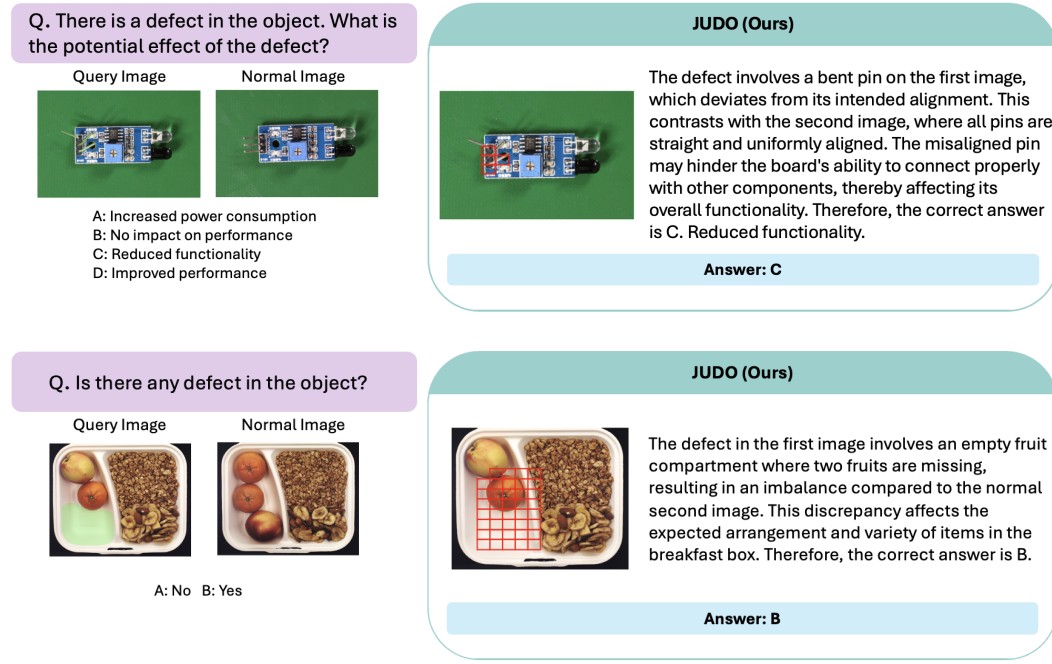

Figure 4: Examples of JUDO's output on the MMAD dataset. The anomalous region in the query image is highlighted in green, while the segmentation output from JUDO is represented as red patches in the image next to JUDO's response.

an incorrect answer ("C"). The Base GRPO + RAG model demonstrates a different failure mode rooted in contextual distraction. Misled by the retrieved domain snippet, which contains descriptions of multiple potential defects, it latches onto a plausible but incorrect description of a "noticeable indentation or dent"—likely related to deformation—instead of the correct "opened" defect type. This failure to ground the textual context in the visual evidence leads to a mislocalization of the "body of the canister" and the wrong answer ("A"). This negative impact is also confirmed by the quantitative results, where the RAG model's accuracy (76.29%) is lower than the Base GRPO model's (77.29%).

In stark contrast, JUDO leverages its internalized knowledge for a successful and coherent analysis. It correctly identifies and locates the defect on the "lid of the canister," provides a visually-grounded explanation of an "irregular white area," and supports its reasoning with an accurate segmentation map to arrive at the correct answer ("D"). This direct comparison clearly illustrates that JUDO's integrated, learning-based approach is essential for achieving reliable and fine-grained reasoning, overcoming the weaknesses of models that either lack domain knowledge or are distracted by it.

## 5 CONCLUSION

In this work, we presented JUDO, a novel framework that addresses the core challenge of aligning domain knowledge in industrial anomaly understanding. By unifying comparative visual reasoning and domain semantics into a domain-aligned learning objective, our approach significantly enhances anomaly understanding capabilities. Experimental results on the MMAD benchmark demonstrate superior performance compared to state-of-the-art models in defect-reasoning tasks. These findings suggest that internalizing domain knowledge and context during training is not only highly effective but also significantly more beneficial than providing context solely at inference time. This approach, therefore, enables more reliable and accurate multimodal reasoning systems in complex industrial scenarios.

## ACKNOWLEDGEMENTS

This work was supported by the Institute of Information and Communications Technology Planning and Evaluation (IITP) and the National Research Foundation of Korea (NRF) through grants funded by the Korean government (RS-2024-00333068, RS-2024-00436936, IITP-2026-RS-2020-II201821, RS-2019-II190421, IITP-2025-RS-2024-00360227, RS-2025-02218768, RS-2025-25442569, RS-2025-24803185, RS-2025-02653113).

## ETHICS STATEMENT

The authors have read and adhered to the ICLR Code of Ethics. This research is based on publicly available datasets for industrial anomaly detection (MMAD, REAL-IAD), which contain images of inanimate industrial objects. No human subjects were involved in this study, and no personally identifiable information was used, thereby minimizing privacy concerns. Our work uses GPT-4o for the programmatic generation of structured training data, as detailed in Appendix A, to ensure a consistent and replicable data construction pipeline and grammar correction. The goal of this research is to advance industrial quality control and automation, and we do not foresee any direct negative societal impacts or ethical concerns arising from the proposed method.

## REPRODUCIBILITY STATEMENT

We are committed to ensuring the reproducibility of our work. The complete source code for our framework, including data preprocessing scripts, implementation of all three training stages, and evaluation protocols, is provided in the supplementary materials, accessible via the repository at `https://github.com/woodavid31/JUDO`. The core methodology of our three-stage training process is detailed in Section 3. Our experimental setup, including the specific datasets, baselines, and key implementation details such as the base model, libraries, and hyperparameters, is described in Section 4.1. Furthermore, a comprehensive description of our programmatic dataset construction pipeline for all training stages can be found in Appendix A. We believe these resources provide the necessary details for the research community to reproduce our results.

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

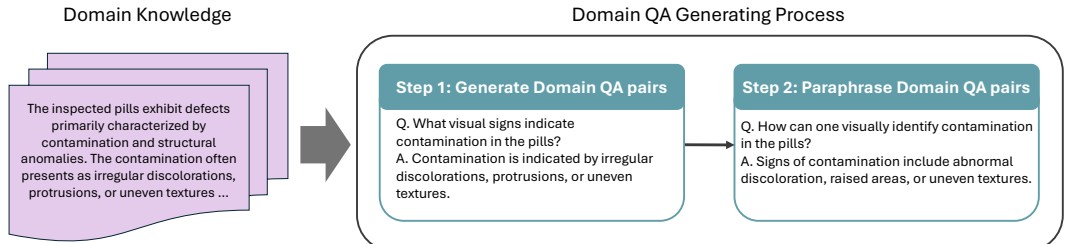

Figure 5: Stage 2 domain Q&A construction pipeline. Illustration of generating domain-specific Q&A pairs from unstructured domain snippets and augmenting them via paraphrasing, using pill contamination defects as an example. The resulting Q&A pairs are paired with normal images to associate textual knowledge with visual context.

## A  DETAILED DATASET CONSTRUCTION PROCESS

We construct the datasets using GPT-4o within a programmatic pipeline. This design ensures high quality, structural consistency, and scalability across all data. The pipeline consists of three stages: constructing comparative explanations (Stage 1), constructing domain Q&A datasets (Stage 2), and generating pseudo-domain reference reasoning (Stage 3). The detailed process for each stage is described below.

**Stage 1: Comparative Explanation.** The primary goal of this stage is to construct a dataset that enables fine-grained comparative reasoning between anomalous and normal images. The construction process for each data instance is as follows:

1. **Input:** Each instance includes a query image containing anomalies, its corresponding binary mask, and a randomly selected normal template image from the same object category.

2. **Anomaly Patch Generation:** The provided anomaly mask is used to identify the precise location of the defect. We overlay a 16×16 grid on the image, and any grid cell overlapping the mask is marked as anomalous. The coordinates of these anomalous patches are then converted into a structured textual sequence. This sequence uses `(row, column)` notation for individual patches and `(row, col_start)-(row, col_end)` for contiguous horizontal patches. This text string is then encapsulated within `<seg>` tags (e.g., `<seg>(11,12)-(11,14), (12,11)</seg>`).

3. **Comparative Explanation Synthesis:** To generate the reasoning text, we first annotate the anomalous query image by using its mask to draw a red outline around the defect region. This visually grounded image, along with the normal template image, the defect type, and any relevant domain notes, is provided as input to GPT-4o. The model is prompted to act as an expert and generate a concise comparative explanation, describing the visual differences between the two images with a focus on the defective region's characteristics. This synthetically generated text forms the content for the `<think>` tags.

4. **Final Data Assembly:** The final training instance combines the visual and textual components. It consists of the original anomalous image and the normal template image as inputs, paired with the generated dual-part response: the `<seg>` tag containing the patch coordinates, followed by the `<think>` tag containing the comparative explanation. Figure 6 shows an example of training data for Stage 1.

**Stage 2: Domain Q&A.** This stage aims to inject category-specific domain knowledge into the model through supervised fine-tuning (Mecklenburg et al., 2024). The dataset is constructed by converting the unstructured textual knowledge from MMAD into a structured Q&A format. The overall process is described in Figure 5.

1. **Input:** We utilize the "domain snippets" from the MMAD benchmark as the foundational knowledge source. Each snippet describes the characteristics of a specific object category and its associated defect types. Table 4 shows an example of a domain snippet for "Squeezed Teeth" defect of the object "Zipper".

Table 3: System prompt for data construction in Stage 2

Generate unique QA pairs grounded in the snippet with enforced categories.
You are tasked with generating high-quality Q&A pairs grounded strictly in the given snippet.
Rules:
- DO NOT mention "snippet" or "according to the snippet".
- All answers must be strictly grounded in the text, no outside knowledge.
- Produce exactly count unique question–answer pairs.
- Use a variety of question styles, including:
1. Criteria-based (e.g., "What criteria indicate a defect in ...?")
2. Defect understanding (e.g., "How does this affect. . . ?")
3. Comparative reasoning (e.g., "What distinguishes X from Y?")
4. Functional impact (e.g., "Why does this defect matter?")
5. Recognition (e.g., "What is. . . ?")
6. Quality control reasoning (e.g., "Why is it important to detect. . . ?")
7. Aesthetic/structural concerns
- Ensure balance: include multiple styles, not just one.
- Keep answers factual and strictly tied to the snippet.
Return as a JSON array: [ {"question": "...", "answer": "..."}]
Snippet:{snippet}

Table 4: Example of a domain snippet for "Squeezed Teeth" defect of the object "Zipper".

The defect manifests as teeth of the zipper that appear squeezed or compressed together, disrupting the otherwise uniform alignment and spacing of the zipper teeth. This irregularity can occur at various locations along the zipper, often concentrated in the central or top/bottom areas. Characteristics of the defect include noticeable distortion in the shape of the teeth, which appear pinched and may vary in spacing compared to the adjacent, properly aligned teeth. The presence of squeezed teeth could potentially hinder the functionality of the zipper, leading to difficulties in smooth opening and closing or causing the zipper slider to get stuck during use. The affected area is identifiable by its misshapen and irregular appearance, contrasting with the uniform and symmetrical pattern of the healthy zipper teeth surrounding it.

2. **Initial Q&A Generation:** For each domain snippet, we employ GPT-4o to generate an initial set of 30 unique question-answer pairs. The generation is guided by a prompt that instructs the model to formulate questions reflecting industrial inspection knowledge, such as inquiries about defect criteria, visual appearance, or functional impact. All answers are strictly grounded in the provided text snippet, without introducing external knowledge.

3. **Paraphrasing for Robustness:** To enhance the model's robustness and prevent overfitting on specific phrasing, each of the initial 30 Q&A pairs is further paraphrased. We use GPT-4o again to generate two semantically identical but lexically diverse paraphrases for each pair. This step expands the dataset by introducing varied phrasings of the same core knowledge.

4. **Final Data Assembly:** The final dataset consists of the original and paraphrased Q&A pairs. To anchor this textual knowledge to a visual context, each Q&A instance is paired with a randomly sampled normal image from the corresponding object category. This encourages the model to associate the domain knowledge with the visual appearance of the object, facilitating better knowledge recall during multimodal reasoning tasks. Figure 7 shows examples of the final Q&A pairs generated from the domain snippets for Stage 2.

# B DATASET EXAMPLES

## B.1 STAGE 1

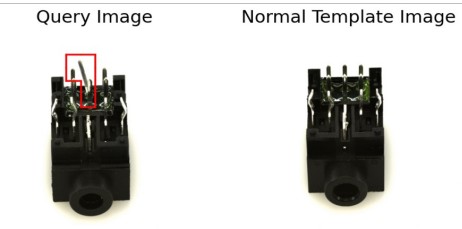

<seg>(3,7)-(3,8), (4,7)-(4,8), (5,8), (6,8)</seg><think>The first image shows a component with a defect where one of the metal pins is bent out of alignment, leaning to one side. In contrast, the second image displays a normal component with all pins straight, indicating proper alignment.</think>

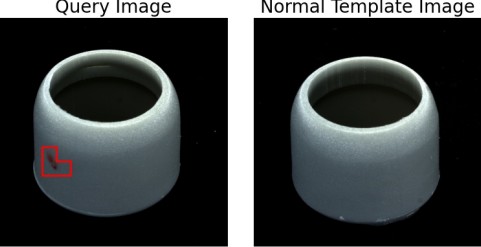

<seg> (10,4), (11,4)-(11,5) </seg> <think> The first image shows a cylindrical object with a defect. This defect appears as a reddish discoloration, suggesting a possible surface damage. In contrast, the second image shows a smooth surface without any irregularities, indicating a normal condition. </think>

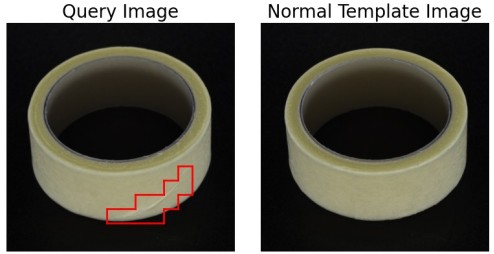

<seg> (11,13), (12,12)-(12,13), (13,10)-(13,12), (14,8)-(14,11) </seg> <think> The first image shows a roll of tape with a defect on its outer surface. The defect appears as an irregular line, disrupting the smooth surface seen in the second image. The second image shows without any irregularities. </think>

Figure 6: Stage 1 training dataset examples

## B.2 STAGE 2

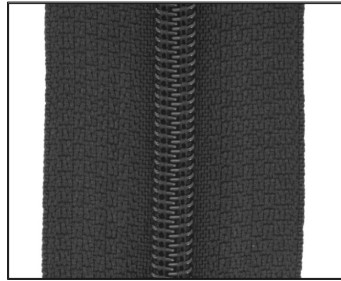

**Original Q:** What might cause fabric border fraying in zippers?
**A:** Fabric border fraying in zippers might be caused
by improper stitching or material degradation.

**Paraphrased Q-1:** What are possible reasons for fabric border fraying
in zippers?
**A:** Possible reasons for fabric border fraying in zippers
include improper stitching or material degradation.

**Paraphrased Q-2:** Why might fabric border fray in a zipper?
**A:** Fabric border might fray in a zipper due to improper stitching
or material degradation.

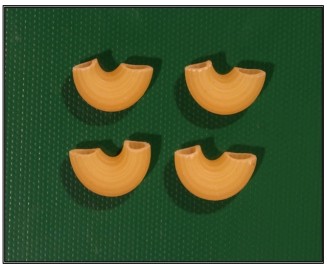

**Original Q:** What ensures the pasta is of standard quality?
**A:** The well-defined ridges on the exterior help indicate
that the pasta is of standard quality.

**Paraphrased Q-1:** How is the quality of the pasta indicated?
**A:** The quality of the pasta is indicated by the well-defined ridges
on its exterior.

**Paraphrased Q-2:** What feature suggests the pasta is standard
quality?
**A:** The feature suggesting the pasta is standard quality
is the well-defined ridges on its exterior.

Figure 7: Stage 2 training dataset examples

# C    ADDITIONAL JUDO RESPONSES

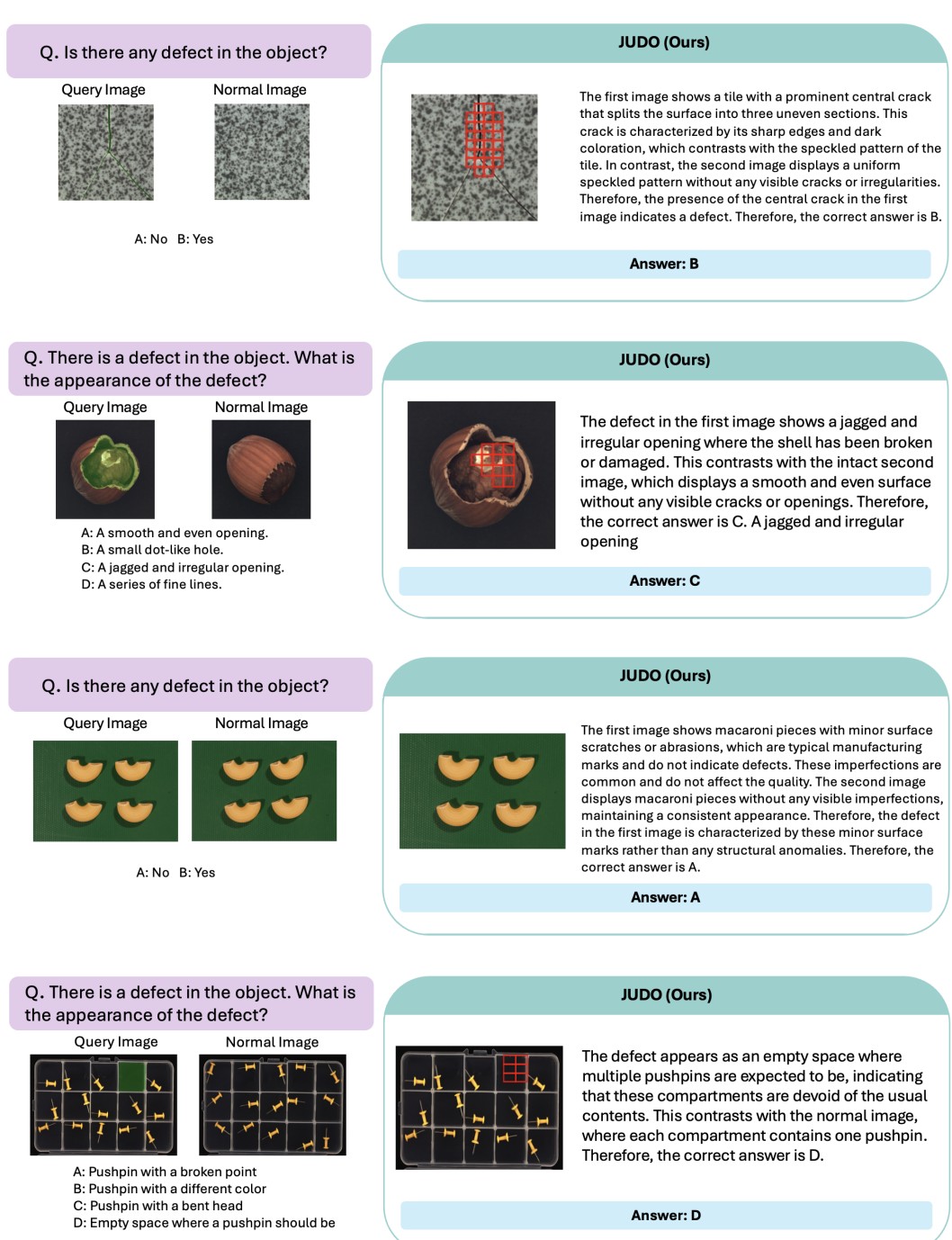

Figure 8: Examples of JUDO's output on the MMAD dataset. The anomalous region in the query image is highlighted in green, while the segmentation output from JUDO is represented as red patches in the segmented image.

