# OpenReview forum: "JUDO: A Juxtaposed Domain-Oriented Multimodal Reasoner for Industrial Anomaly QA"
_ICLR.cc/2026/Conference — ICLR 2026 Poster_

### Official Review · Reviewer_EjVp · 2025-10-18

**Soundness:** 3
**Presentation:** 3
**Contribution:** 2
**Rating:** 6
**Confidence:** 4

**Summary:**

This paper focuses on multi-modal anomaly reasoning via large-scale multi-modal language model. Specifically, it considers how to make the multi-modal language model be more aware of the domain-specific anomaly. To achieve this, it proposes a three-stage model training/post-training pipeline by considering the anomaly discrimination of normal and abnormal cases, the supervised instruction tuning and reinforcement-learning-based post training. The proposed method acgieves state of the art perfroance of anomaly reasoning on the MMAD benchmark.

**Strengths:**

- The proposed method is well-motivated, seeking to enhance domain-specific anomaly understanding by injecting fine-grained anomaly knowledge throughout the training stages of a large multi-modal language model.

- The proposed three-stage training pipeline, which integrates both visual and linguistic anomaly data, is logically sound and effectively develops strong anomaly reasoning capabilities.

- The reward engineering implemented in the reinforcement-based post-training stage is well-designed and appropriate for the task.

- The paper is well-written and easy to follow, presenting a clear and logical flow of ideas.

**Weaknesses:**

- The methodological contribution appears limited, as the proposed framework primarily involves a sequential application of standard techniques (instruction-based fine-tuning and reinforcement learning-based post-training). While effective, this approach offers fewer novel insights into the fundamental challenges of injecting anomaly knowledge into multi-modal language models.

- The proposed method demonstrates a significant performance degradation in anomaly discrimination compared to the Qwen-2.5 baseline. The paper offers several hypotheses for this outcome in the analysis section, but these are not experimentally validated, which undermines their credibility. A more rigorous analysis, such as an ablation study, is needed to clarify which training stage introduces the apparent trade-off between anomaly detection and anomaly reasoning capabilities.

- The manuscript lacks a detailed discussion of the training data division and statistics, which are critical for evaluating the effectiveness of the post-training stage. Furthermore, the paper's central claim—that injecting domain knowledge during training is superior to doing so at test-time—is not substantiated with experimental evidence. A direct comparison is needed to validate this assertion.

- The comparison to general-purpose LLMs could be strengthened. While the inclusion of GPT-4o is a good start, this baseline is rapidly becoming dated. To provide a more current and comprehensive understanding of the performance landscape, it would be beneficial to include comparisons with more recent state-of-the-art commercial models (e.g., Claude 3, Gemini 1.5).

**Questions:**

Please see the weaknesses.

---

> ### Author Response · Authors · 2025-11-21
> **Response to Reviewer EjVp (1/2)**
>
> We appreciate your insightful comments and constructive feedback. We address each concern in detail below.
>
> ---
>
> ## W1. Overall Novelty
>
> We highlight that JUDO’s novelty lies in unifying comparative visual cues and domain semantics into one domain-aligned reasoning process, rather than treating them as isolated components. Stage 1 introduces juxtaposed segmentation, enabling the model to learn patch-level comparisons between defect and normal images—unlike prior systems that localize defects without using paired contextual information. Stage 2 internalizes domain semantics by transforming raw defect descriptions into structured supervision, in contrast to approaches that rely on prompts or RAG to inject domain knowledge externally.
>
> Stage 3 integrates these two capabilities through GRPO with domain-oriented rewards covering reasoning quality, segmentation alignment, and answer accuracy. Prior GRPO pipelines rely only on accuracy or generic reasoning rewards and do not incorporate domain semantics. JUDO therefore goes beyond a standard SFT–RL pipeline by aligning visual grounding and domain knowledge within a single training process, which is key for robust and reliable industrial anomaly QA.
>
> ---
>
> ## W2. Ablation Study on the Training Stages
>
> Thank you for the comment. We added an empirical analysis to verify the source of the discrimination drop. In the revised manuscript, we have added Figure 2, where we report anomaly discrimination accuracy and averaged defect-reasoning accuracy (defect classification, localization, description, and analysis) across all checkpoints (baseline without CoT, baseline with CoT, Stage 1, Stage 1+2, and the full JUDO model). The results in Figure 2 show that the largest drop occurs when CoT is enabled in the baseline model, before any JUDO training, indicating that the decline is driven by a reasoning-induced behavioral shift rather than our pipeline. Within the JUDO pipeline, anomaly detection accuracy fluctuates slightly while defect reasoning  performance increases steadily across stages. This demonstrates that the stages in JUDO improve reasoning without causing a major anomaly discrimination drop.
>
> [1] Section 4.2 in Mind Your Step (by Step): Chain-of-Thought can Reduce Performance on Tasks where Thinking Makes Humans Worse, ICML 2025.

---

> ### Author Response · Authors · 2025-11-21
> **Response to Reviewer EjVp (2/2)**
>
> ## W3-1. Training Data Division and Statistics
> The training data consists of 1.4k Real-IAD (cross domain) images and 293 MMAD images (~3\% of MMAD, sampled as one per category). We additionally generate ~13k domain QA pairs for Stage 2 using MMAD knowledge snippets. For Stage 3 GRPO training, we use 1.4K MMAD QA instances corresponding to the same 293 images, yielding roughly 4–5 questions per image, without introducing any additional MMAD samples.
>
> ---
>
> ## W3-2. Experiment on Comparing Training-Time and Test-Time Domain Knowledge Injection
>
> We would like to clarify that the requested comparison is provided in Table 2. The row “+ GRPO + RAG” corresponds to test-time knowledge injection via retrieval-augmented generation, achieving 76.29\% accuracy. In contrast, the row “+ GRPO + DomInj” corresponds to training-time domain knowledge injection, which achieves 79.82\% accuracy — a +3.5\% improvement. This directly supports our claim that internalizing domain knowledge during training is more effective than relying on test-time retrieval.
>
> ---
>
> ## W4. More Recent State-of-the-Art Commercial Models as Baseline
>
> We evaluate JUDO against recent multimodal commercial models such as Gemini-2.5-Pro, Gemini-2.5-Flash, and GPT-5-Mini. Accordingly, we have updated the main Table 1 in the manuscript.
>
> | Model            | Anomaly Detection | Defect Reasoning Tasks |
> |------------------|-------------------|---------------------------|
> | Gemini-2.5-pro   | 83.07             | 76.82                     |
> | Gemini-2.5-flash | 93.38             | 72.79                    |
> | Gemini-2.5-flash + CoT | 83.99  | 51.06         |
> | Gpt-5-mini       | 64.09             | 75.54                   |
> | JUDO             | 64.51             | 80.06                     |
>
> The above table reports the average accuracy across four MMAD defect-reasoning subtasks (defect classification, localization, description, and analysis). Gemini-2.5-Pro and Gemini-2.5-Flash achieve strong performance on anomaly detection, whereas defect-reasoning tasks remain comparatively weak. We interpret the reason for Gemini's superior performance in anomaly detection as (1) its state-of-the-art vision encoder, (2) under a no-CoT setting where vision-dominant tasks often perform better without CoT, as we have explained in W2. Despite these advantages, Gemini still underperforms JUDO on defect-reasoning subtasks, showing that a small, open-source model that is domain-aligned can outperform much larger commercial models when the task requires specialized industrial reasoning.
>
> We note that using CoT for Gemini-2.5-flash does not improve the performance; both anomaly detection and defect-reasoning tasks show worse performance, especially in a large margin for defect-reasoning tasks.

---

### Official Review · Reviewer_mhqc · 2025-10-26

**Soundness:** 2
**Presentation:** 2
**Contribution:** 2
**Rating:** 2
**Confidence:** 3

**Summary:**

This paper proposes JUDO, a multimodal reasoning framework for industrial anomaly understanding. The method integrates visual comparative segmentation, domain knowledge injection via supervised QA fine-tuning, and domain-oriented GRPO alignment to improve both defect localization and explanation quality.

**Strengths:**

1. The problem is well-motivated, addressing the domain knowledge gap that limits existing VLMs in industrial anomaly analysis.
2. The three-stage training pipeline is clear, systematic, and practically meaningful. The incorporation of juxtaposed segmentation grounding enhances fine-grained visual reasoning, while domain QA fine-tuning contributes to more reliable textual analysis.
3. The paper is clearly structured and the experimental results are presented in a well-organized manner.

**Weaknesses:**

1. The proposed framework is mainly an integration of existing components—comparative segmentation, supervised domain knowledge tuning, and GRPO-based post-training—and thus reads more as pipeline engineering rather than introducing a fundamentally new reasoning mechanism or architectural innovation. The methodological novelty therefore appears limited.
2. The model depends on GPT-4o–generated reasoning chains as pseudo ground truth, yet the paper does not evaluate whether these rationales are reliable, consistent, or susceptible to hallucination. This creates a conceptual gap, since the paper also claims that general LMMs (including GPT models) lack domain reasoning capability.
3. The paper attributes the performance degradation in anomaly discrimination to catastrophic forgetting but provides no empirical analysis or mitigation. Without verification or intervention, this explanation remains speculative and raises concerns about the robustness of the multi-stage training pipeline.

**Questions:**

1. My interpretation is that the method essentially fine-tunes an existing VLM on domain-specific multimodal data, achieving better performance due to domain alignment. Could the authors clarify what they consider the core novelty beyond a well-designed training pipeline?
2. The paper argues that GPT-based multimodal models lack sufficient industrial anomaly reasoning capability. However, GPT-4o is still used as the source of "golden reasoning" in the training pipeline. How do the authors justify the reliability and consistency of GPT-generated reasoning as ground truth in a domain where GPT is claimed to be insufficient? More specifically, how do you ensure the reasoning produced is correct and not hallucinated?

If I have misunderstood the intended contribution or positioning, I would appreciate clarification from the authors.

---

> ### Author Response · Authors · 2025-11-21
> **Response to Reviews mhqc (1/2)**
>
> We appreciate your insightful comments and constructive feedback. We address each concern in detail below.
>
> ---
>
> ## W1. Methodological Novelty
>
> We would like to take this opportunity to emphasize the methodological novelty. The main novelty lies not in integrating existing modules but in establishing a unified domain alignment in industrial domain understanding. Each stage contributes an essential domain alignment that differs from existing LMM. We summarize the novelty of each stage as follows.
>
>
> **Stage 1 - Juxtaposed segmentation for visual grounding:**
>
> We juxtapose query images against normal images for patch-level segmentation. Prior anomaly QA methods typically predict regions independently without comparative reasoning tied to grounded evidence. Also, instead of treating the normal image as optional context, JUDO incorporates it as a core reasoning context during training.
>
> **Stage 2 - Domain-knowledge injection:**
>
> Unlike prior work that injects textual domain knowledge externally via prompts or RAG, we directly internalize domain semantics into model parameters, yielding more reliable domain-oriented reasoning. To achieve this, we transform the raw, unstructured defect descriptions into a structured supervision format that the model can effectively learn from.
>
> **Stage 3 - Domain-oriented GRPO:**
>
> We unify the visual grounding and domain semantics using GRPO-based optimization under our proposed rewards regarding (1) domain alignment reasoning, (2) anomaly segmentation, and (3) accurate and formatted decision-making. This differs from prior GRPO-based methods, which only rely on simple accuracy or heuristic reasoning rewards.
>
> Overall, our framework goes beyond a standard SFT–RL pipeline by aligning visual grounding, domain context, and reasoning within a unified training process, which is essential for robust industrial anomaly QA.
>
> ---
>
> ## W2, Q2. Using GPT-Generated Reasoning as Ground Truth
>
> We clarify that GPT-4o is not used as a domain expert nor as a source of new industrial knowledge. In our pipeline, GPT-4o receives (1) ground-truth labels, (2) the relevant portion of the domain snippet, (3) comparative cues from the Stage 1 pipeline using ground-truth segmentation, and its role is constrained to structuring these into a coherent rationale rather than performing open-ended inference. Thus, GPT functions as an evidence organizer, a widely acknowledged capability [1], rather than a source of novel reasoning. To mitigate hallucination, we strictly constrain the rationale to the provided evidence without inferring new knowledge.
>
> We now refer to these outputs as **pseudo-domain rationales** rather than golden rationales to avoid implying expert-validated industrial reasoning. Their purpose is solely to organize the provided evidence into a structured explanation. Importantly, these rationales are not directly supervised; they are used only as semantic alignment rewards. Also, this differs from typical GRPO variants that optimize with shallow correctness signals such as answer matching or token-frequency heuristics, rather than rewarding reasoning quality. We have updated the terminology and added clarification in Section 3.3 to prevent misinterpretation.
>
> [1] "We Need Structured Output": Towards User-centered Constraints on Large Language Model Output, CHI 2024.

---

> ### Author Response · Authors · 2025-11-21
> **Response to Reviews mhqc (2/2)**
>
> ## W3. Discussion about the Degradation of Anomaly Discrimination in a Multi-Stage Training Pipeline
>
> We have expanded our analysis and added Figure 2, which tracks performance across the base model (Qwen2.5-VL), the same model with chain of thought enabled, and each component of our pipeline. The figure shows that the main drop in anomaly detection occurs before our training begins, when the model shifts from direct answering to a reasoning-oriented mode. This shift reduces anomaly detection from 71.39\% to 61.90\% and also lowers the averaged accuracy across the four defect subtasks (classification, localization, description, and analysis) from 65.15\% to 57.00\%. These results indicate that the degradation originates from the shift to reasoning rather than from forgetting. During our staged learning, anomaly detection remains stable, with only a small fluctuation from 65.42\% after Stage 1+2 to 64.51\% after Stage 3, while the four defect subtasks improve significantly across all stages.
>
> Taken together, these findings show that the drop in anomaly discrimination performance is caused primarily by the shift to a reasoning mode, and the multi-stage training itself remains stable while substantially enhancing the reasoning-dependent defect subtasks.
>
> ---
>
> ## Q1. Core Novelty Beyond Well-Designed Training Pipeline
>
> We clarify that JUDO goes beyond tuning a pretrained VLM with domain-specific data. The main claim of JUDO is that domain knowledge and contextual information fundamentally matter in industrial anomaly QA, yet this approach remains underexplored and non-trivial. In this direction, our core contribution is a unified learning objective in which comparative visual reasoning and domain semantics fuse together into the domain-aligned reasoning process rather than from a sequence of independent processing stages. This unified framework is conducted using only a single anomaly sample per defect type in MMAD, rather than memorizing specific patterns. We believe this integrated objective constitutes a methodological contribution beyond a simply well-engineered pipeline.

---

### Official Review · Reviewer_ixi3 · 2025-10-30

**Soundness:** 3
**Presentation:** 4
**Contribution:** 3
**Rating:** 6
**Confidence:** 4

**Summary:**

This paper presents JUDO, a Juxtaposed Domain-Oriented Multimodal Reasoner, to address the current weaknesses in industrial anomaly detection using LMMs (Large Multimodal Models). While the GRPO-based LMMs are optimized for tasks grounded in general knowledge, pinpointing the defects in industrial setups requires superior domain knowledge and visual context through a three-stage progressive framework. The framework uses juxtaposed segmentation learning, supervised fine-tuning (SFT) that injects domain-specific knowledge, and reinforcement learning (GRPO) with tailored multi-component rewards for domain-aware reasoning, resulting in an average accuracy of 80.73% and improvement in explainability through domain-aligned, visually grounded reasoning.

**Strengths:**

1. The three-stage learning framework proposed in the paper is logical and helps advance the integration of domain knowledge into multimodal reasoning. The three-stage progressive learning pipeline (SegJux, DomInj, GRPOdom) effectively combines visual segmentation, textual domain knowledge, and reinforcement learning. This hierarchical design is both conceptually strong and technically sound, marking a meaningful step beyond conventional GRPO-based post-training methods.

2. The visual grounding via segmentation and the domain-aligned textual reasoning make outputs transparent and practical for industrial use. By segmenting the input images into semantically meaningful regions (e.g., defect areas, product components, the model anchors its subsequent textual explanations to specific visual evidence. This ensures that every reasoning step is traceable to an observable region within the image, reducing errors and improving trustworthiness.

3. This paper addresses a significant industrial challenge of creating an automated, explainable anomaly detection system. The findings have potential applicability across various industry segments like manufacturing, logistics, etc.

4. State-of-the-art average on MMAD versus both general LMMs and AnomalyR1; ablations demonstrate additive benefits of each stage.

**Weaknesses:**

1. Compared with other general-purpose models, JUDO lags in binary anomaly detection tasks. JUDO’s score (64.51%) in simple anomaly-vs-normal classification tasks is lower than that of general-purpose VLMs such as Kimi-VL (72.93%). Although the paper claims this can be attributed to the advanced vision encoder of Kimi-VL, this could also suggest that the model’s complex reasoning optimization may not translate well to simpler binary tasks.

2. Reasoning mode not systematically evaluated: The paper argues that free-form CoT can hurt MMAD accuracy, thus opting for a constrained `<seg>/<think>/<answer>`  format with reward shaping. However, the trade-offs among no reasoning, free CoT, and structured reasoning are not empirically studied. A controlled comparison would clarify why structured reasoning helps and when CoT hurts, strengthening claims.

3. The paper does acknowledge that sequential training across three stages can cause performance degradation in earlier-learned tasks. Although addressed to some extent, this issue remains a known weakness of multi-phase fine-tuning and may affect scalability to more diverse industrial domains.

**Questions:**

1. Could the authors provide an ablation comparing different reasoning styles – (a) direct answer (no reasoning), (b) free-form Chain-of-Thought, and the proposed structured `<seg>/<think>/<answer>` reasoning – to quantify the trade-offs between interpretability and task accuracy?

2. JUDO underperforms compared to general-purpose models in binary anomaly discrimination. Could the authors clarify whether it is due to catastrophic forgetting, visual encoder limitations, or reinforcement objectives prioritizing reasoning quality over detection precision?

---

> ### Author Response · Authors · 2025-11-21
> **Response to Reviews ixi3 (1/2)**
>
> We appreciate your insightful comments and constructive feedback. We address each concern in detail below.
>
> ---
>
> ## W1. JUDO Lags in Binary Anomaly Detection
> We agree with the reviewer that the result is mixed, especially for binary anomaly detection, which relies heavily on low-level visual cues. While Kimi-VL’s strong performance can be partly explained by its more capable vision encoder, we agree that it is not the only contributing factor.
>
> | Model        | Scale | Direct Answering           | Reasoned Answering      |
> |--------------|-------|------------------|----------|
> | Qwen2.5-VL   | 7B    | 71.39           | 61.92    |
> | Kimi-VL-A3B  | 16B   | 72.93           | 61.30    |                  |
> | JUDO         | 7B    | 61.32           | 64.51    |
>
> ---
>
> The table above shows the anomaly-detection performances under direct answering and reasoned answering settings. Both Qwen and Kimi show a substantial drop when moving from direct answering to reasoning-based prompts. In contrast, JUDO maintains stable performance under reasoning, indicating that our optimization mitigates this vulnerability. This result aligns with prior findings that CoT can reduce performance on vision-heavy tasks—an effect observed not only in industrial anomaly detection [1] but also in broader vision benchmarks [2]. Since this phenomenon is directly related to W2, we further discuss in the next section, where we present the full ablation.
>
> [1] MMAD: A Comprehensive Benchmark for Multimodal Large Language Models in Industrial Anomaly Detection, ICLR 2025.
>
> [2] The Illusion of Thinking: Understanding the Strengths and Limitations of Reasoning Models via the Lens of Problem Complexity
> Authors, Neurips 2025.
>
> ---
>
> ## W2, Q1. Experiments on Different Reasoning Styles
>
> We thank the reviewer for pointing out the need for a systematic comparison. In the tables below, we include an ablation evaluating (a) No reasoning, (b) free-form CoT, and \(c) our structured `<seg>` `<think>` `<answer>` reasoning across Qwen2.5-VL, Kimi-VL, and JUDO.
>
> > **Accuracy on the Anomaly Discrimination Task**
>
> | Model        | Scale | No Reasoning | Free-form CoT | Structured Reasoning |
> |--------------|-------|------------------|----------|-----------------------|
> | Qwen2.5-VL   | 7B    | 71.39           | 61.92    | 60.54                |
> | Kimi-VL-A3B  | 16B   | 72.93           | 61.30    | 55.02                    |
> | JUDO         | 7B    | 61.32           | 52.27    | 64.51                |
>
> > **Average Accuracy across Other Tasks**
>
> | Model        | Scale | No Reasoning | Free-form CoT | Structured Reasoning |
> |--------------|-------|------------------|----------|-----------------------|
> | Qwen2.5-VL   | 7B    | 72.75           | 66.75    | 61.65                |
> | Kimi-VL-A3B  | 16B   | 74.18           | 53.77    | 44.41                    |
> | JUDO         | 7B    | 82.65           | 81.14    | 83.42
>
> Across both anomaly discrimination and other tasks except the anomaly detection task, free-form CoT consistently reduces accuracy for Qwen and Kimi, confirming that CoT can harm performance on vision-dominant tasks. Prior work reports the same effect: when tasks rely on fine-grained perceptual cues, verbal reasoning interferes with non-verbal processing, which is a phenomenon known as verbal overshadowing [3]. This explains the sharp performance drop observed for general-purpose VLMs under free CoT.
>
> Structured reasoning leads to an even larger drop for Qwen and Kimi. Here, the performance degrades further because, in addition to the drawback of verbal reasoning, these models are not aligned to follow multi-part constrained formats. In contrast, JUDO improves under structured reasoning because it is explicitly trained to generate coherent `<seg>` evidence, `<think>` reasoning, and `<answer>` predictions.
>
> Overall, the ablation demonstrates that
> (1) Free CoT can be detrimental to visual tasks, and
> (2) Structured reasoning helps only when the model is optimized for it, as in JUDO.
> These findings directly address the reviewer’s concerns and support our choice of structured reasoning.
>
> [3] Section 4.2 in Mind Your Step (by Step): Chain-of-Thought can Reduce Performance on Tasks where Thinking Makes Humans Worse, ICML 2025.

---

> ### Author Response · Authors · 2025-11-21
> **Response to Reviews ixi3 (2/2)**
>
> ## W3, Q2. Regarding Multi-Stage Optimization Stability and the JUDO’s Performance on Binary Anomaly Discrimination.
>
> We thank the reviewer for raising these important questions regarding multi-stage stability (W3) and JUDO’s performance on binary anomaly discrimination (Q2). We have added Figure 2 to the manuscript, which visualizes the performance progression from the Qwen2.5-VL baseline, baseline with CoT, and each stage of JUDO. This figure makes clear that the dominant source of performance degradation comes from the shift from direct answering to a reasoning mode: anomaly detection drops sharply from 71.39% to 61.90%, and the average accuracy over the four defect subtasks (defect classification, localization, description, and analysis) decreases from 65.15% to 57.00%, even before any of our optimization begins. This reasoning-induced effect aligns with our earlier observations in our response. In contrast, across our three optimization stages, anomaly detection remains largely stable, with only a small fluctuation from 65.42% after Stage 1+2 to 64.51% after Stage 3, indicating that forgetting exists but is very minor.
>
> Figure 2 also highlights that the main effect of our staged learning is a substantial improvement in the four defect subtasks: the averaged accuracy increases from 62.58% (Stage 1) to 67.18% (Stage 1+2), and ultimately to 80.57% in JUDO. This demonstrates that our GRPO-based structured reasoning alignment effectively strengthens JUDO’s defect-understanding ability while keeping anomaly detection stable. The remaining gap in binary anomaly discrimination is primarily due to the nature of the task, which relies heavily on low-level visual cues and benefits little from improved reasoning; thus, models with stronger visual encoders naturally achieve higher scores on this specific metric.
>
> Taken together, the updated results show that (1) the shift to reasoning mode itself is the main source of the initial degradation in anomaly detection, (2) multi-stage learning introduces only minimal forgetting, and (3) JUDO achieves large improvements where reasoning is required while maintaining stable anomaly-detection performance. We have revised the manuscript accordingly and included Figure 2 for clarity.

---

### Official Review · Reviewer_wrJU · 2025-11-01

**Soundness:** 3
**Presentation:** 2
**Contribution:** 3
**Rating:** 4
**Confidence:** 3

**Summary:**

1. The paper presents JUDO (Juxtaposed Domain-Oriented Multi-modal Reasoner), a framework designed to enhance industrial anomaly detection by integrating domain-specific knowledge and contextual reasoning into large multimodal models (LMMs).
2. JUDO performs visual comparative reasoning by juxtaposing defect and normal images for fine-grained inspection, and injects domain expertise through supervised fine-tuning followed by reinforcement learning (GRPO) with tailored rewards.
3. Experiments on the MMAD benchmark show that JUDO outperforms strong baselines such as Qwen2.5-VL-7B and GPT-4o, highlighting the value of domain-oriented reasoning for anomaly understanding.

**Strengths:**

1. Novel Multistage Training Framework:
The paper introduces a well-structured two-stage pipeline combining supervised fine-tuning (SFT) and reinforcement learning (GRPO), effectively aligning domain-specific knowledge with multimodal reasoning objectives.

2. Conceptual Contribution in Knowledge Integration:
It provides a meaningful conceptual advancement in how to inject out-of-domain or domain-specific knowledge into large multimodal language models, enabling more accurate and interpretable reasoning for industrial anomaly understanding.

**Weaknesses:**

1. Clarity and Presentation:
The paper’s presentation is somewhat difficult to follow, particularly in the implementation and dataset construction sections.
For example, in Lines 216–224, it would be helpful to include a concrete example comparing a QA pair derived directly from the domain snippet versus one generated for general inspection knowledge, to clearly illustrate their differences.

2. Incomplete Baseline Comparison:
The experimental section lacks evaluation against recent state-of-the-art vision–language models (VLMs) such as Gemini 2.5 Pro, which would provide a stronger and more convincing benchmark for JUDO’s effectiveness.

**Questions:**

1. Dataset Scale and Composition:
How many QA pairs were generated for the “general inspection knowledge” subset? It would be useful to know the relative proportion of these conceptual QA pairs compared to those derived directly from domain snippets in MMAD.

2. Use of External Knowledge in Closed-Source Models:
Given that open-source VLMs still lag slightly behind but demonstrate strong reasoning capabilities, could the proposed “general inspection knowledge” QA corpus be used as external prompt knowledge for closed-source models such as GPT-4o, GPT-5, or Gemini 2.5 Pro?
How would this prompt-based knowledge injection compare to the performance of the proposed fine-tuned JUDO model in terms of reasoning accuracy and interpretability?

---

> ### Author Response · Authors · 2025-11-21
> **Response to Reviews wrJU (1/2)**
>
> We appreciate your insightful comments and constructive feedback. We address each concern in detail below.
>
> ---
> ## W1. Clarity of Dataset Construction in Stage 2
>
> Thank you for the helpful suggestion. We would like to clarify that all QA pairs used in Stage 2 are generated strictly grounded on the raw domain snippets provided by MMAD, and these correspond to inspection knowledge derived from the snippets. Because the domain snippet is unstructured descriptive text rather than question–answer content, QA pairs cannot be directly extracted from it. We understand that our previous wording might be misleading as general inspection knowledge referred to a separate subset. Accordingly, we have revised the dataset construction paragraph in Section 3.2 to avoid this ambiguity.
>
> In addition, we have added new tables in the Appendix (Table 3 and 4) that provide the prompt for generation and a concrete example of a domain snippet that illustrates how it forms the basis for generating the QA pairs. We believe these revisions improve the clarity and presentation of the Stage 2 data construction process.
>
> ---
>
> ## Q1. Dataset scale and Composition
> As addressed in W1, all Stage-2 QA pairs are generated from the raw domain snippet for each category, and we do not maintain separate subsets for general inspection knowledge versus snippet-derived QA pairs. The domain snippet is unstructured descriptive text without question–answer content, so all QA pairs are produced by prompting GPT-4o to reflect the inspection knowledge implied in the snippet.
>
> For each category, we generate 30 base QA pairs and paraphrase each twice, resulting in roughly 90 QA pairs per category and approximately 13k QA pairs in total. Because all QA pairs are generated from the snippets through the data construction pipeline, there is no distinct proportion to report across different QA types.

---

> ### Author Response · Authors · 2025-11-21
> **Response to Reviews wrJU (2/2)**
>
> ## Q2. Use of External Knowledge in Closed-Source Models.
>
> Our QA corpus is not suitable for direct prompt injection because it was constructed as a training signal—it contains paraphrased and redundant QA pairs optimized for supervision, not for clean declarative reference. Moreover, selecting “relevant” QA pairs from this corpus for a specific image would already require knowing the defect label, which makes it unusable as neutral external guidance.
>
> | Model          | Anomaly Detection | Defect Reasoning Tasks |
> |----------------|-------------------|----------------------------|
> | GPT-4o         | 68.63             | 69.51                     |
> | GPT-4o + RAG   | 65.69             | 70.29                     |
> | JUDO  | 64.51             |               80.06      |
>
>
> To fairly evaluate prompt-based knowledge injection, we instead use a RAG setup with concise domain descriptions. As the Table shows, adding RAG to GPT-4o lowers anomaly-detection accuracy and provides only a small improvement in the averaged accuracy in defect reasoning tasks. The averaged accuracy is calculated over the four MMAD defect-reasoning subtasks (defect classification, localization, description, and analysis), where JUDO still outperforms GPT-4o by a large margin. This is because models without internal domain grounding tend to rely heavily on external prompts, whereas JUDO learns the domain knowledge directly. As a result, JUDO is both more accurate and more efficient than prompt-based knowledge injection in defect-reasoning subtasks that require reliable domain-oriented reasoning.
>
> ---
>
> ## W2. Incomplete Baseline Comparison
>
> We have additionally evaluated JUDO against more recent multimodal commercial models such as Gemini-2.5-Pro, Gemini-2.5-Flash, and GPT-5-Mini, and revised the main table in the manuscript.
> | Model            | Anomaly Detection | Defect Reasoning Tasks |
> |------------------|-------------------|---------------------------|
> | Gemini-2.5-pro   | 83.07             | 76.82                     |
> | Gemini-2.5-flash | 93.38             | 72.79                    |
> | Gemini-2.5-flash + CoT | 83.99  | 51.06         |
> | Gpt-5-mini       | 64.09             | 75.54                   |
> | JUDO             | 64.51             | 80.06                     |
>
> Gemini-2.5-Pro and Gemini-2.5-Flash achieve strong performance on anomaly detection, whereas tasks requiring domain-level reasoning remain comparatively weak. We interpret the reason for Gemini's superior performance in anomaly detection lies in (1) its state-of-the-art vision encoder, which excels at fine-grained visual cues, (2) evaluation under a no-CoT setting where vision-dominant tasks often perform better without explicit reasoning.
> Despite these advantages, Gemini-2.5 variants in overall still underperform JUDO on defect-reasoning tasks, showing that a small, open-source model that is domain-aligned can outperform much larger commercial models when the task requires specialized industrial reasoning.
>
> As additional information, using CoT for Gemini-2.5-flash degrades performance in anomaly detection and especially in a large margin for defect-reasoning tasks.

---

### Author Response · Authors · 2025-12-02
**Official Comment by Authors**

We sincerely appreciate the reviewers’ time, effort, and constructive insights. We have carefully addressed all questions and concerns in our individual responses and have updated the PDF accordingly. All modifications are highlighted in red in the revised PDF.

Since further discussion is not possible at this stage, we briefly summarize (1) our key contributions and strengths identified by reviewers, and (2) the primary concerns raised by the reviewers.

---

# Contributions and Strengths

- **Core contribution**: JUDO is the first systematic approach to internalize domain knowledge into both visual and textual reasoning in industrial anomaly understanding under one-shot settings. Unlike prior models, we claim that domain knowledge and contextual information fundamentally matter for industrial anomaly QA, yet this direction remains underexplored and technically non-trivial.

- **Reviewer acknowledgment of novelty and design**: Our contribution on aligning domain-specific knowledge is acknowledged by Reviewer wrJU, and the overall design is recognized as technically sound or practically meaningful by Reviewer ixi3, Reviewer mhqc, and Reviewer EjVp.

- **Strengths identified by reviewers**: Furthermore, the reviewers have pointed out the strengths of each stage: (1) Juxtaposed segmentation enhances fine-grained visual reasoning, as noted by Reviewer mhqc; (2) Injecting domain knowledge into LMMs enables more accurate reasoning and reliable textual analysis, as noted by Reviewers wrJU and mhqc; (3) GRPO with our domain reasoning and segmentation reward unifies domain semantics and visual grounding, which Reviewer EjVp highlights as well-designed and appropriate for the task.

- **Superior performance**: As a result, JUDO achieves state-of-the-art overall performance (80.73%) among open-source models and fine-tuned models like AnomalyR1 (77.29%) used as baselines. Furthermore, JUDO also demonstrates strong domain-reasoning capability (80.06%) in tasks such as defect classification, localization, description, and analysis, outperforming Gemini-2.5-Pro (76.82%) in reasoning-required settings.
---

# Primary Concerns and our Responses

&nbsp;

### **1. Clarity of Stage 2 dataset construction and composition *(wrJU)***

We clarified that Stage-2 QA pairs come exclusively from MMAD domain snippets and not from multiple knowledge sources. We updated Section 3.2 with a clearer construction description and included concrete examples in the appendix.

---

### **2. Novelty beyond standard SFT-RL pipeline  *(mhqc, EjVp)***

We emphasized that the novelty does not lie in simply combining existing components but in establishing a unified domain alignment framework for industrial anomaly understanding. Each stage contributes to domain alignment in a complementary way, leading to behaviour that individual parts cannot achieve alone.

---

### **3. Anomaly detection performance drop & multi-stage stability  *(ixi3, EjVp, mhqc)***

We added a discussion in the manuscript with Figure 2 showing that the anomaly detection drop comes from enabling reasoning in the base model, rather than from the training stages itself. In addition, anomaly detection performance remains stable across the training stages, while defect-reasoning capability improves substantially.

---

### **4. Reasoning-style ablation *(ixi3)***

We conducted an ablation, evaluating (a) No reasoning, (b) free-form CoT, and \(c) our structured \<seg> \<think> \<answer> reasoning across Qwen2.5-VL, Kimi-VL, and JUDO. We show that free-form CoT decreases performance for all models, that structured reasoning leads to further degradation for Qwen/Kimi, and that JUDO instead improves under the same structured format due to GRPO-aligned reasoning.

---

### **5. Comparisons with more recent commercial models  *(wrJU, EjVp)***

We added Gemini-2.5-Pro/Flash and GPT-5-Mini to the main table, which perform strongly in anomaly detection but still fall short in defect reasoning compared to JUDO.

---

### **6. Using GPT-4o to generate ground-truth rationale  *(mhqc)***

We clarified that GPT-4o is not used as a domain expert and only organizes the evidence that is already provided into pseudo-domain rationales. And these are used only as semantic reward signals rather than direct supervision.

---

### Meta-Review · Area_Chair_eh47 · 2026-01-11

**Summary:**

This paper introduces JUDO, an industrial anomaly detection framework that efficiently incorporates domain knowledge and context in visual and text reasoning. Reviewers find this work technically sound and practically meaningful. Several concerns were raised about novelty, multi-stage stability, anomaly detection performance drop, and comparisons with more recent commercial models. While the novelty is not that significant, most of these issues were adequately addressed during the rebuttal. Overall, the strengths of the work outweigh its weaknesses.

**Reviewer Concerns:**

Reviewer wrJU

W1: Clarity of dataset construction. [addressed by the rebuttal]

W2: Incomplete baseline comparison. [addressed by the rebuttal]

---
Reviewer ixi3

W1: Experiments on different reasoning styles. [addressed by the rebuttal]

W2: Multi-stage optimization stability. [in the middle]

R2: Although the authors explain through current experimental results, the scalability to more diverse industrial domains is not fully addressed.

---
Reviewer mhqc

W1: Lack of novelty. [in the middle]

R1: There is some novelty, but not significant.

W2: GPT-generated reasoning as ground truth. [in the middle]

R2: The response reduced conceptual confusion, but there is no empirical evaluation of the rationale quality.

W3: Performance degradation in anomaly discrimination. [addressed by the rebuttal]

---
Reviewer EjVp

W1: The methodological contribution appears limited. [in the middle]

R1: There is some novelty, but not significant.

W2: Ablation study on the training stages. [addressed by the rebuttal]

W3: Training data division and statistics. [addressed by the rebuttal]

W4: Comparing training-time and test-time domain knowledge injection. [addressed by the rebuttal]

**Reviewer Scores:**

Reviewer wrJU: 4 -> 6 (all concerns are addressed, raises the score)

Reviewer ixi3: 6 -> 6 (remains a positive rating)

Reviewer mhqc: 2 -> 4 (a part of the concerns are addressed)

Reviewer EjVp: 6 -> 6 (remains a positive rating)

Average score: 5.5

---

### Decision · Program_Chairs · 2026-01-26

Accept (Poster)